

# Subaqueous speleothems as archives of groundwater recharge on Australia's southern arid margin

Calla N. Gould-Whaley[1], Russell N. Drysdale[1], Pauline C. Treble[2,3], Jan-Hendrik May[1], Stacey Priestley[4,]
John C. Hellstrom[1], Clare Buswell[5]

[1]School of Geography, Earth and Atmospheric Science, University of Melbourne, Parkville, 3010, Australia
[2]Australian Nuclear Science and Technology Organisation, Lucas Heights, 2234, Australia
[3]School of Biological, Earth and Environmental Sciences, University of New South Wales, Kensington, 2052, Australia
[4]Drought Resilience Mission, Commonwealth Scientific and Industrial Research Organisation, Adelaide, 5000, Australia
[5]College of Humanities, Arts and Social Sciences, Flinders University, Adelaide, 5001, Australia

*Correspondence to*: Calla N. Gould-Whaley (cgould@student.unimelb.edu.au)

**Abstract.**

As anthropogenic climate change enhances aridity across vast regions of the globe, understanding drivers of aridification is more important than ever before. Unfortunately, arid regions globally tend to exhibit a paucity of palaeoclimate records, and the archives that are available typically comprise unconsolidated sediments prone to reworking, large dating uncertainties, and ambiguous climatic interpretations. This is certainly true of Australia's vast continental interior, which is dominated by harsh, arid conditions. Mairs Cave, in the southern Ikara-Flinders Ranges (South Australia), is located on the southern margin of the arid zone. In the present day the cave is largely dry and there is limited evidence of active speleothem growth. However, historical records and observations throughout the cave indicate that it has been periodically flooded, suggesting the local water balance was once much more positive than it is today. The cave contains a curtain of hanging speleothems known as pendulites, which grow subaqueously when submerged in water that is saturated with respect to calcite. Geochemical evidence, including trace element concentrations, uranium isotope ratios, and Dead Carbon Fractions, all indicate that rising of the local groundwater during periods of enhanced groundwater recharge is the cause of the cave flooding events that trigger pendulite growth. Uranium-thorium dating of a pendulite retrieved from Mairs Cave has revealed two multi-millennial growth phases (68.5 to 65.4 kyr and 51.2 to 42.3 kyr) and two short bursts of growth (18.9 kyr and 16.4 kyr) during the Last Glacial Period. The absence of subsequent pendulite growth suggests that strong water deficits under warm Holocene interglacial conditions give rise to episodic, rather than persistent, cave flooding.

## 1 Introduction

Spatial and temporal gaps in records of past climate limit our understanding of climate dynamics and variability, thus hindering the development of rigorous models that can predict the heterogenous manifestations and magnitudes of anthropogenic climate change (Skinner, 2008). Arid regions across the globe are underrepresented by palaeoclimate records compared to humid regions due to environmental conditions that are not conducive to the capture nor the preservation of palaeoclimate information



(Falster et al., 2018; Fujioka and Chappell, 2010; Habeck-Fardy and Nanson, 2014). Moisture deficits preclude the existence of perennial lakes, bogs, and fens, and limit the abundance of flora and fauna necessary to generate fossil records (Bateman et

al., 2007; Carr et al., 2007; Field et al., 2018; Reeves et al., 2013b). Additionally, biological material tends to be poorly preserved in arid environments due to desiccation, photodegradation, oxidation, extreme temperature fluctuations, and physical weathering by abrasive sediments (Carr et al., 2007; Field et al., 2018; Fujioka and Chappell, 2010; Scott, 2016; Thomas and Burrough, 2016). Aeolian archives, while abundant in arid regions, can be difficult to interpret given that modern analogues and mathematically modelled aeolian landscapes exhibit complex and non-linear responses to climate drivers (Fitzsimmons et

al., 2013; Telfer et al., 2017). Palaeolacustrine and palaeofluvial archives are composed of unconsolidated sediments that can be re-worked by strong winds, and the records contained within these archives are often discontinuous, and typically underpinned by dating methods that can yield large uncertainties (Fitzsimmons et al., 2013; Telfer et al., 2017).

Australia is no exception to the global pattern; the vast interior of the continent is dominated by harsh, dry conditions and is

consequently lacking in palaeoclimate records (Reeves et al., 2013b). Given that Australia is the largest landmass in the Southern Hemisphere, terrestrial records of Australian palaeoclimate are essential to understanding global climate change and variability. A higher density of records in coastal regions allows for detailed, continuous, and well-dated reconstructions of past climate along the fringes of the continent (Cadd et al., 2021; Reeves et al., 2013a, b; Williams et al., 2009). However, these findings cannot be extrapolated to the arid interior. Kati Thanda-Lake Eyre, the ephemerally filled endorheic salt lake at

the heart of the continent, has thus far been a focus of much palaeoclimate research in arid Australia (Cohen et al., 2022; DeVogel et al., 2004; Fu et al., 2017; Magee et al., 1995, 2004; May et al., 2022; Nanson et al., 1998). Beach ridges and other lacustrine landforms and sediments provide indisputable evidence of lake-filling events. However, the 1 200 000 km$^2$ catchment responsible for filling the lake sits largely to the north and receives rainfall from the Indo-Australian Summer Monsoon (IASM; Pook et al., 2014). Therefore, records of lake filling reflect hydroclimatic changes in the northern part of the

continent rather than the climate of the interior itself.

Munda-Lake Frome is another endorheic salt lake 400 km SE of Kati Thanda. Inflow to Munda is primarily received as runoff from the Ikara-Flinders Ranges to the west, but lake filling can also occur if monsoonal rainfall in the northern reaches of the continent is especially high (May et al., 2015). Between Munda and Kati Thanda there is a string of smaller lakes (Malakanha,

Blanche, and Gregory). During periods of especially high recharge Munda, Malakanha, Blanche, and Gregory can overflow into one another and become a hydrologically connected mega-lake (mega-Munda; Cohen et al., 2012, 2015; May et al., 2015). There is also evidence to suggest that during the most extreme pluvial periods of the LGP, mega-Munda and Kati Thanda became connected via overflow through the Warrawoocara channel (Cohen et al., 2011, 2012; Leon and Cohen, 2012). While these lake systems provide indisputable evidence of large-scale environmental change, dating uncertainties still limit the utility

of these sedimentary archives to detect climate events at millennial and centennial timescales through the Last Glacial Period



(LGP) and beyond. As a result, our knowledge of the climatic past of Australia's arid interior remains fragmented both spatially and temporally.

Speleothems, secondary calcium carbonate deposits that form in caves, contain physical and chemical properties that can act
as proxies of past climate change (Fairchild and Baker, 2012). Since speleothems are dateable by radiometric methods, the proxy records can be tethered to precise chronologies to produce climatic reconstructions with high temporal resolution, sub-annually in some cases (e.g. Faraji et al., 2024; Lu et al., 2021; Mattey et al., 2008; Van Rampelbergh et al., 2015). Speleothems with slower growth rates can provide exceedingly long, continuous records of past climate (e.g. Drysdale et al., 2020; Hodge et al., 2008; Ünal-Imer et al., 2016). Due to their subterranean setting, speleothems are shielded from surface processes capable
of degrading, truncating, or eroding other terrestrial archives (Fairchild and Baker, 2012). While post-depositional alteration can occur, it is usually detectable, and affected specimens can be excluded from climate reconstructions (Bajo et al., 2016; Frisia, 2015; Martínez-Aguirre et al., 2019; Martín-García et al., 2014). Speleothems develop in carbonate bedrock terrains across the globe (Atsawawaranunt et al., 2018; Comas-Bru et al., 2020) and consequently have been employed to address a wide range of palaeoclimatic questions (e.g. Cheng et al., 2016; Corrick et al., 2020; Cruz et al., 2009; Dumitru et al., 2019,
2021; Vaks et al., 2013; Wang et al., 2008). Speleothems typically form when water drips through the ceiling of the cave, which means that (for at least part of the year) precipitation must exceed evapotranspiration such that water can infiltrate the subsurface (Fairchild and Baker, 2012). Therefore, encountering a speleothem-bearing cave in a present-day arid environment generally suggests that at some point in the past the local water balance was more positive. These speleothems can contain records of hydroclimate before and/or during the onset of aridification, which have never been more important than in our
current era of anthropogenic climate change, which threatens enhanced aridity in many regions of the globe (Feng and Fu, 2013; Markowska et al., 2016, 2020).

Mairs Cave, in the central Ikara-Flinders Ranges, is situated at the southernmost tip of the Kati Thanda-Lake Eyre basin, and on the boundary between the arid zone to the north and the semi-arid zone to the south (Fig. 1). The site is also at the interface
between tropical and mid-latitudinal climate systems, receiving moisture from both the Southern Hemisphere Westerly Winds (SHWWs) and the IASM (Pook et al., 2014). Mairs Cave has been the subject of one previous palaeoclimate study (Treble et al., 2017), where regional palaeohydrology was reconstructed from two stalagmites that together spanned the Last Glacial Maximum (LGM) and early deglaciation. Stalagmites are the preferred speleothem type for past climate reconstructions as they often exhibit thick and flat laminae along their central axis, which makes for easy sampling and high-resolution records
(Fairchild and Baker, 2012). There are few stalagmites in Mairs Cave; to circumvent this issue, we chose to explore the potential of alternative formations as paleoclimate archives.

Mairs Cave contains a curtain of bulbous speleothems suspended from the ceiling that exhibit features characteristic of subaqueous calcite growth. These unique speleothem formations are known as "pendulites" - formations that begin as



stalactites then become submerged in water that is supersaturated with respect to calcite resulting in subaqueous deposition of calcite on the existing stalactite surface (Boop et al., 2014). The initial phase of subaqueous growth can seal off the original path of dripping water that fed the stalactite, such that pendulite growth can only continue under subaqueous conditions. Hiatuses in subaqueous growth occur when water levels drop below the pendulite, or when the water is no longer supersaturated with respect to calcium carbonate (Boop et al., 2017; Dumitru et al., 2021; Fornós et al., 2002; Onac et al., 2022; De Waele et

al., 2017). In this study, we present morphological, microstratigraphic, and geochemical evidence to demonstrate that pendulite growth in Mairs Cave corresponds to intervals of higher local water table during periods of enhanced groundwater recharge. We conclude that these speleothems constitute important archives of past hydrological change in Australia's southern arid zone where palaeoclimate records are lacking.

## 2 Site Description

Mairs Cave (32.18 °S, 138.87 °E) is on Adnyamathanha Country, in South Australia (Fig. 1). The site receives between 77.4 and 582.8 mm/year, with a mean of 290.4 mm/year (1917 to 2023; BoM, 2024a). Mean potential evapotranspiration is 1720 mm/year, making the region one

of extreme water deficit (Aridity Index = 0.17; arid classification; Arora, 2002; Burrell et al., 2020). Winter rainfall is primarily delivered to the study site by low pressure systems carried across the Southern Ocean by the SHWWs (Meneghini et al., 2007). Throughout the year,

tropical moisture can be delivered to the region when moist air over the Indian Ocean moves poleward to create a Northwest Cloudband (NWCB) across Australia (Reid et al., 2020). Tropical moisture delivery can be enhanced during summer by southward propagation of depressions associated

with the IASM (Dey et al., 2021; Pook et al., 2014; Reid et al., 2020). Positioned within reach of both tropical and mid-latitudinal climate systems, palaeoclimate archives from Mairs Cave thus have potential for tracking their influence on rainfall in Southern Australia beyond the instrumental era.


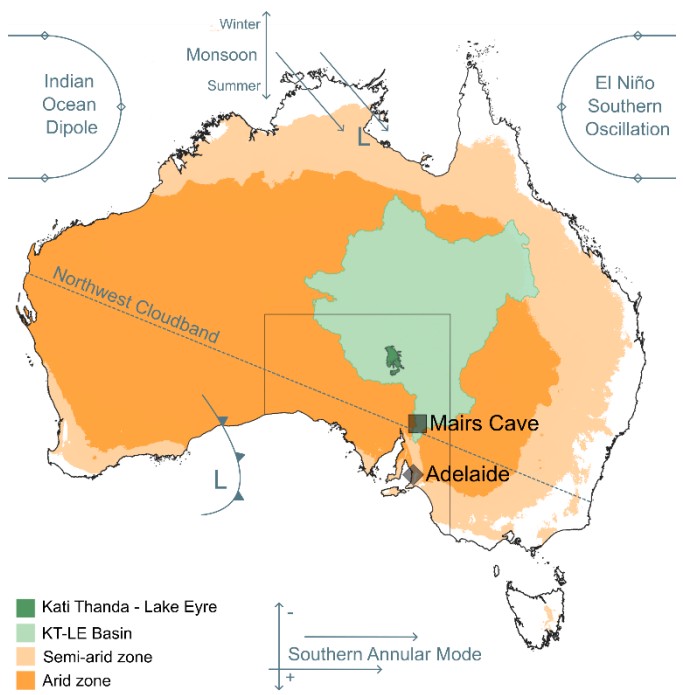

**Figure 1  Location of Mairs Cave (study site) relative to Adelaide, (the capital city of the state of South Australia). Location of Kati Thanda-Lake Eyre and the extent of the KT-LE Basin. Regions designated arid (0.05 < AI < 0.2), regions designated as semi-arid (0.2 < AI < 0.5; Trabucco and Zomer et al. 2022). Also depicted are the primary systems that influence climate at the study site.**



Although there are many cave-bearing carbonate rock formations scattered across the Ikara-Flinders Ranges, the largest and most decorated caves, including Mairs, tend to occur in the Etina Formation (~655 Ma) which is a sandy, oolitic and stromatolitic limestone inter-bedded with siltstones (Fromhold and Wallace, 2012; Lawrence, 2009).  Mairs Cave is situated in a strike ridge of Etina limestone that rises 16 m above the floodplain of Buckalowie Creek (Fig. 2a), an ephemeral waterway

that runs parallel to the Etina outcrop in a NNE direction (Fig. 2b). Mairs Cave appears to have formed along the boundary between two distinct facies within the limestone, one easily weathered (the eastern wall), the other less soluble (the western wall). The cave is accessed via a 17 m vertical shaft formed within the more weatherable facies. The base of the entrance shaft opens out to the main chamber of the cave (Fig. 2c), which is ~100 m long and 12 m wide. Beyond the main chamber, the cave narrows and continues for an additional 120 m in a northeast direction, following the strike of the Etina limestone. Cave

temperature measured by data loggers in 2019 ranges between 19.9 and 15.3 °C in the main chamber (open to the surface via the vertical entrance shaft) with a mean annual temperature of 18.0 °C. Temperatures are more stable in the interior of the cave, ranging between 21.8 and 21.4 °C, with an annual mean of 21.6 °C. Over the same period, daily surface air temperatures fluctuated between 44.6 and 6.7 °C, with a mean of 24.5 °C.

The western wall of the main chamber is near vertical and is thinly coated in calcite to a height of ~3 m, suggestive of standing water (5 in Fig. 2d; Fig. 3a). Large breakdown blocks fallen from the cave ceiling are coated with thin layers of calcite (4 in Fig. 2d; Fig. 3b and 3c). The eastern wall is well-decorated with an assemblage of hanging formations, many of which have a bulbous, cauliflower-like surface morphology, suggestive of subaqueous overgrowth (2 in Fig. 2c and 2d; Fig. 3e). Throughout the cave, the floor and lower shelves are strewn with 'calcite rafts', which are thin plates of calcite that typically precipitate at

the surface of a waterbody that is supersaturated with respect to calcium carbonate (Fig. 3d). Speleothem rubble is scattered throughout the cave but is mostly cemented to the cave floor by calcite coatings. Collectively, these observations indicate that, despite the present arid conditions, the chamber was once at least partially filled with water supersaturated with respect to calcite. Stalagmites MC-S1 and MC-S2 studied by Treble et al. (2017) were retrieved from the main chamber (6 in Fig. 2c and 2d) and a side passage off the main chamber (3 in Fig. 2c and 2d). MC-S2 contains layers of calcified sediment, most likely

deposited when floodwater mobilised sediments on the cave floor. MC-S1 exhibits no visible sediment layers, presumably because it sat higher in the cave and on top of a boulder, meaning it was not within reach of the sediment on the cave floor.

The cave is effectively dry today. Some formations appear damp but not actively dripping, precluding a dripwater monitoring study. Reports in the South Australian Department of Mines Annual Review noted that guano mining efforts in the cave had

been hindered by a deep pool of standing water in the main chamber (Winton, 1920, 1922). In the archived newsletters of the Caving Exploration Group of South Australia (1956-present, available from the Australian Speleological Society at https://st1.asflib.net/JNS/SA/CEGSA/CEGSA-Pubs.html), three trip reports mention the presence of standing water in Mairs Cave. Since cave flooding is a noteworthy event, we assume there to have been no water within the cave if no mention of water is made in a trip report. Flooding of the cave was observed in June of 1968. A trip report from three months prior made no





mention of water in the cave. In June 1974, there was approximately 3 m of water in the bottom of the shaft. Cave divers noted that the water was "crystal clear". A trip in December of the same year reported water at the base of the entrance shaft 15 cm deep. In addition to these three trips where water was observed within the cave, 44 trip reports either noted the absence of water or made no mention of water.

**Figure 2** **(a) Cross section of the Buckalowie Valley indicating the position of Mairs Cave, Buckalowie Creek and the elevations of the relevant features referred to in the text. (b) Aerial view of Buckalowie Valley showing the geological formations present. The dashed line indicates the position of the cross section in a. Distance from cave entrance can be read from a. (c) Cross section of Mairs Cave showing the elevations of the relevant features referred to in the text (extracted from a 3D model of the cave). (d) Aerial map of the cave (extracted from 3D model of the cave, cross-checked against the original map published by the Cave Exploration Group of South Australia, with surveying and mapping carried out by Sexton et al. 1958). Distance from cave entrance can be read from c.**



Despite near total dependence on groundwater resources, the hydrogeology of the Ikara-Flinders Ranges has not been extensively studied (Fildes et al., 2020). No hydrogeological studies have taken place in Buckalowie Valley, but findings from

elsewhere in the region suggest that ridges of resistant formations tend to create hydrological divides between aquifers hosted in more fractured and/or porous formations (Ahmed et al., 2021; Ahmed and Clark, 2016; Fildes et al., 2020). Buckalowie valley is constrained on either side by siltstone aquitards with extremely low primary porosity (Yankaninna, Uroonda, Tarcowie, and Ulupa Siltstones in Fig. 2a and 2b), therefore it is likely that the boundaries of the aquifer roughly align with the topographic boundaries of the valley. There are only 5 bores within the valley, for which there is limited data available and

no long-term monitoring. The lack of data precludes potentiometric surface mapping or hydrogeological modelling to establish the flow direction, recharge/discharge areas, or aquifer volume. In 1980 a bore (Unit number 6633-230; Fig. 2b) was drilled into the Etina Limestone 0.9 km NNE of the cave entrance. The standing water level at the time was 5.54 meters below the surface (Fig. 2a). In 2018 the original bore was sealed, and a new bore (Unit number 6633-660; Fig. 2b) was drilled 50 m to the east to facilitate transition to a solar-powered pump. Given the proximity of the bore, the age of the Etina Limestone (>

635 Ma; Giddings et al., 2009) and the hydrological connectivity generally observed in mature karst landscapes (Ford and Williams, 2013; LeGrand, 1983; LeGrand and Stringfield, 1971),  it is quite likely that the aquifer from which the bore draws water is continuous along the entire length of the limestone outcrop.





Figure 3  Photos taken in Mairs Cave. (a) calcite precipitated on the western wall of the main chamber to a height of 3 m. (b) breakdown blocks in the main chamber. (c) small breakdown block in the main chamber that has been cleft to display calcite overgrowth. (d) calcite rafts in an alcove along the passage past the main chamber.  (e) curtain of bulbous formations suspended from the ceiling of an overhang on the eastern wall of the main chamber.






# 3 Methods and Materials

## 3.1 Pendulite sampling and analyses

We collected a small speleothem (MC19; 4 x 10 cm) from a discrete position behind a curtain of larger formations, ensuring its removal was neither visible nor damaging to surrounding formations. The specimen was halved using a diamond masonry blade, set in epoxy resin for sampling, and polished to reveal the internal stratigraphy. To identify potential changes in deposition state a thin section was prepared from one half for petrographic examination, which was carried out using a Leica DM750 Polarisation Microscope. For uranium-thorium dating, powders of approximately 10 mg were milled from laminae before and after suspected hiatuses, using a New Wave Micromill. Uranium-thorium (U-Th) measurements were made on a Nu Plasma Multi-Collector Inductively Coupled Plasma Mass Spectrometer (MC-ICP-MS) in the Isotope Geochemistry Laboratory at the University of Melbourne using the protocols of Hellstrom (2003, 2006). Ages were corrected with an initial thorium ($^{230}Th/^{232}Th$) activity ratio of $1.5 \pm 1.5$.

Cation concentrations (Mg, Ca, Sr) in MC19 and NIST-610 glass were measured using a Helex laser-ablation system with a Lambda Physik Compex 110 ArF excimer laser (193 nm) coupled to an Agilent 7700x quadrupole ICP-MS (Drysdale et al., 2012; Jochum et al., 2011). The 8.5 mm-long scan line crossed the boundary from the internal stalactite to the suspected subaqueous material, with a pre-ablation spot size of 60 mm and data-collection spot size of 26 mm. Pre-ablation was performed with a laser pulse rate of 15 Hz, which was reduced to 10 Hz for data collection. Scan velocity was set to 13 mm s-1 and laser fluence to 30 J cm-2.

## 3.2 Calcite raft sampling and analyses

Calcite rafts on the floor of the main chamber were covered in a layer of fine sediment. A single dusty raft (MCCR-33) was collected. About 10 mg of the dusty exterior was removed with a scalpel and reserved for U-Th isotopic analysis (MCCR-33_1). From the exposed surface another 10 mg was removed (MCCR-33_2), followed by another 10 mg (MCCR-33_3) until visibly clean calcite remained, from which a final 10 mg were sampled (MCCR-33_4). These four samples with decreasing degrees of dust contamination underwent U-Th isotopic analysis to better constrain the initial thorium required for correcting the U-Th ages of the calcite rafts. U-Th analysis of the calcite raft samples occurred according to the same procedure as the pendulite samples.

Rafts on ledges and in hollows along the cave wall were generally cleaner, thicker, and exhibited larger crystals. Four clean calcite rafts (MCCR-25, 26, 30, 31) were collected for paired U-Th and radiocarbon analysis. From each raft a chip (15 mg) underwent U-Th isotopic analysis. The remainder of each sample was reacted with 85% $H_3PO_4$ at room temperature. $CO_2$ evolved from the outer portion of each sample (ca. 12% by weight) was pumped away to avoid sample surface contamination





with atmospheric $CO_2$. The sample was then transferred to a hot block at 90 °C and the $CO_2$ that evolved over a 1-hour period was converted to graphite using the $Fe/H_2$ method (Hua et al., 2001). Accelerator Mass Spectrometry (AMS) $^{14}C$ measurement was carried out using the at the Australian Nuclear Science and Technology Organisation (ANSTO) VEGA Facility (Wilcken et al., 2015). After correction for machine background, procedural blank and isotopic fractionation using measured $\delta^{13}C$, $^{14}C$

content was reported as percent modern carbon (pMC) and radiocarbon age (Stuiver and Polach, 1977). The Dead Carbon Fraction (DCF) was calculated for each of these samples according to Equation 1 (Hua et al., 2012):

$$DCF = \left(1 - \frac{A_{calcite}}{A_{atm}}\right)100\%$$ (1)

Where $A_{calcite}$ is the measured radiocarbon content of the calcite rafts (pMC) and $A_{atm}$ is the radiocarbon content of the atmosphere (in pMC; from SHcal20, Hogg et al., 2020) at the time of calcite precipitation (established by U-Th dating).

An additional seven clean calcite rafts (MCCR-10, 16, 18, 20, 22, 29) were collected from the cave and ground into fine powders and homogenised. Subsamples of 0.5 mg were used for measuring cation concentrations. The remaining powder

underwent U-Th dating. Mg/Ca and Sr/Ca ratios in the calcite rafts were determined by digesting 0.5 mg subsamples in 5 mL of a 3% Supelco Suprapur $HNO_3$ solution. Measurements were carried out at ANSTO using a Varian Vista-PRO Simultaneous Inductively Coupled Plasma Atomic Emission Spectrometer (ICP-AES) according to the procedure described by Carilli et al. (2014) and ANSTO Method I-3775.

**3.3 Groundwater sampling and analyses**

Groundwater was sampled from the active bore closest to the cave (Unit number 6633-660) in 2019, 2021, and 2022. The pump was left to purge for 30 minutes, equivalent to three times the volume of standing water within the bore. A Terumo 50 mL Luer lock syringe was rinsed three times with bore water then a Millex filter unit with a 0.45 µm low-binding Millipore membrane was fitted to the syringe. The first 150 mL passing through the filter were discarded. HDPE collection bottles (acid

washed prior to fieldwork) were rinsed with the filtrate three times before being filled to the brim and capped. Care was taken to leave no headspace in any of the bottles. After being firmly sealed, the lids and necks of the bottles were wrapped in parafilm. A Supelco Suprapur 65% $HNO_3$ spike was added to one of the blanks and to the bore water samples intended for major and minor cation analysis to reduce the pH to below 2. Upon returning to the laboratory all samples were stored at 4 °C.

All groundwater analyses were performed at ANSTO. The concentrations of the major cations (Mg, Ca, and Sr) were measured on a Thermo Fisher iCAP7600 ICP-AES (ANSTO Method I-3775). The concentration of U was measured using a Varian 820MS ICP-MS (ANSTO method VI2809). Trace element to calcium ratios ($X/Ca_{ppt}$) in calcites that could theoretically



precipitate from these groundwaters were calculated according to Equation 2 using the measured groundwater ratios ($X/Ca_{soln}$) and a range of partition coefficients ($D_X$) available in the literature (Day and Henderson, 2013; Drysdale et al.,

2019; Fairchild et al., 2000; Huang et al., 2001; Huang and Fairchild, 2001; Tremaine and Froelich, 2013):

$$D_X = \frac{X/Ca_{ppt}}{X/Ca_{soln}} \qquad\qquad (2)$$

On the groundwater sample collected in 2022 the concentration of major anions (Cl⁻, $SO_4^{2-}$, Br-, and $NO^{3-}$) was measured

using a Dionex DX-600 ion chromatograph (ANSTO method I-5279). Saturation indices for calcite and other minerals were calculated using the WATEQ4F thermodynamic database in PHREEQC Interactive 3.7.3.15968 program (Parkhurst and Appelo, 1999).

To constrain groundwater age, the tritium concentration was measured with a Perkin Elmer Quantulus ultra-low level Liquid

Scintillation Analyser (LSA) following the procedures described by Morgenstern and Taylor (2009). Radiocarbon dating of the same water by AMS was performed according to the methods described in Wilcken et al. (2015). The radiocarbon age was rounded according to Stuiver and Polach (1977). The Han and Plummer closed-system model was used to correct the groundwater radiocarbon age to account for geochemical processes that occur along the flow path (Han and Plummer, 2013). The radiocarbon ages of groundwater were converted to calendar years before present (1950) using a calibration curve (Bard

et al., 1998; Fairbanks et al., 2005; Kitagawa and van der Plicht, 1998; Plummer et al., 2004; Plummer and Glynn, 2013; Reimer et al., 2013; Stuiver et al., 1998). For comparison to the calcite rafts, geochemical adjustment models were used to estimate the DCF of the groundwater DIC i.e. what portion of the carbon present in the groundwater sample was contributed by exchange with the bedrock or other 'dead' carbon reservoirs (Han and Plummer, 2013; Kalin, 2000).

## 4 Results

### 4.1 Physical characteristics of MC19

The stratigraphy of MC19 can be macroscopically divided into two distinct facies. Inside the blue line in Figure 4, the older, interior portion comprises white to pale yellow calcite with regions of well-defined laminae, interspersed with blurry, poorly defined laminae and clear, featureless calcite. This internal portion exhibits both vertical and horizontal extension. These features as well as the fact that calcite growth is emanating from the upper section that was attached to the cave ceiling, clearly

suggest this portion of MC19 is a composite stalactite exposed in long section. Outside of the blue line in Figure 4, the exterior growth is beige/brown in colour and clearly laminated throughout with a distinctive radial growth pattern. This composite structure is typical of a 'pendulite', a speleothem that begins as a stalactite in an air-filled cave and becomes submerged in a body of water that is supersaturated with respect to calcite, at which point the original stalactite acts as a nucleus upon which



calcite is subaqueously deposited for as long as the pendulite remains
submerged and the water remains supersaturated with respect to calcite (Boop
et al., 2014).

The transition from the stalactite to the subaqueous material can also be
identified in thin section (Fig. 5a). Flat stalactite laminae are crosscut by thick,
rounded, micritic layers. This angular unconformity suggests a dissolution
event (e.g. via condensation corrosion) occurred prior to the subaqueous
deposition of calcite. The subaqueous material is primarily composed of
composite crystals interrupted by layers of clotted peloidal micrite (M in Fig.
5b) and microsparite (Ms in Fig. 5b). We interpret the thickest and darkest
micritic layers as corresponding to extended growth hiatuses and stretches free
of micrite correspond to periods of uninterrupted growth i.e. sustained flooding
of the cave. By extension, clustered yet distinct micrite layers may be
indicative of fitful growth due to intermittent cave flooding.

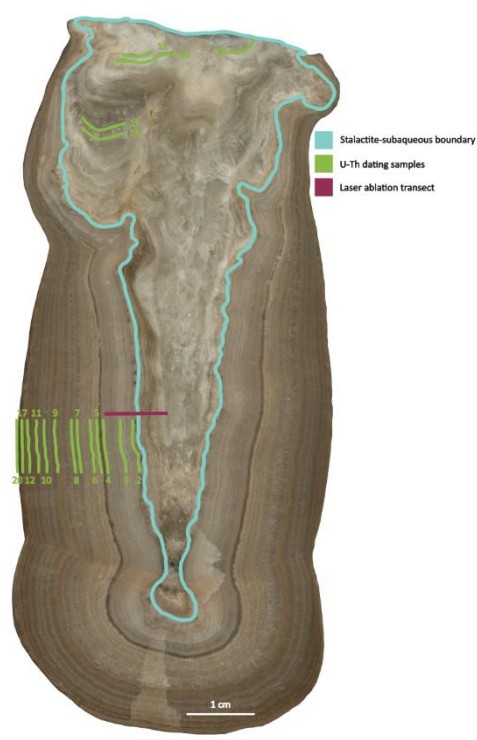

**Figure 4  MC19 in vertical cross section, with the boundary between the internal stalactite and the external subaqueous overgrowth indicated by the blue line, the laser ablation transect, and the U-Th sampling.**

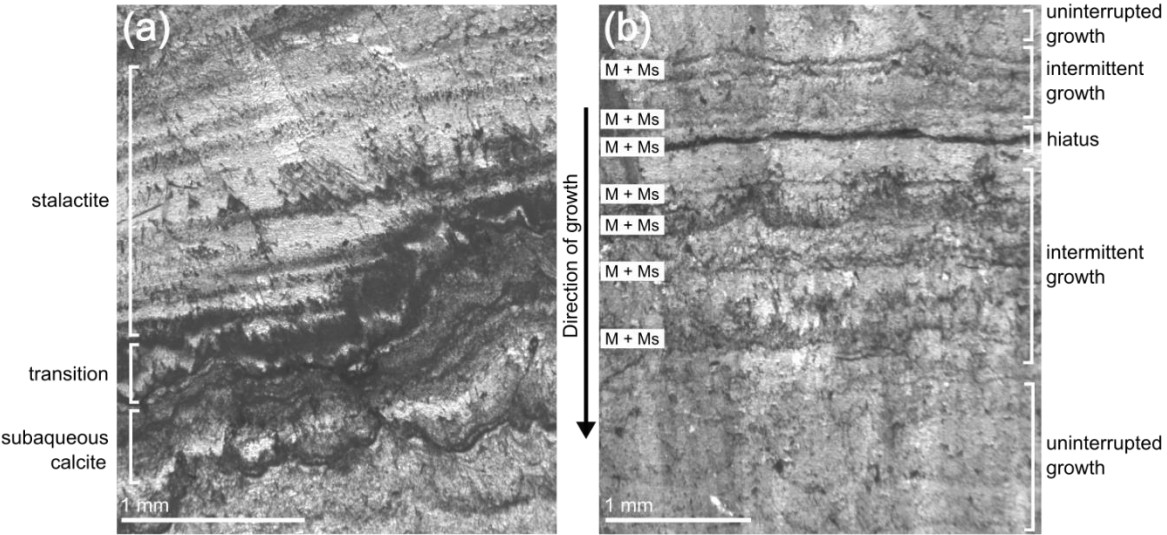

**Figure 5  Thin section images of MC19 under plane-polarised light. (a) at the transition between the stalactite and the subaqueous overgrowth. (b) the most visually prominent layer of micrite and microsparite in the subaqueous overgrowth.**





## 4.2 Trace elements

Trace element concentrations in MC19 were measured along a transect that traversed the boundary between the internal stalactite and the subaqueous overgrowth (yellow line in Fig. 4). There were abrupt increases in all trace elements when
crossing into the subaqueously deposited calcite. For example, mean Mg/Ca values increased from 7.70 mmol/mol in the stalactite to 36.25 mmol/mol in the subaqueous material. Mean Sr/Ca increased from 0.30 to 0.71 mmol/mol. In the calcite rafts, average Mg/Ca was 38.00 mmol/mol and average Sr/Ca was 0.87 mmol/mol. Groundwater cation concentrations measured in samples collected in 2019, 2021, and 2022 are shown in Table 1. Mg/Ca was 0.93, 1.02, and 1.00 mol/mol in the three years respectively. Sr/Ca was 5.42, 5.63, and 5.77 mmol/mol.


**Table 1   Cation concentrations measured in groundwater.**

|  | Mg (mmol/L) | Sr (μmol /L) | Ca (mmol/L) |
|---|---|---|---|
| Groundwater sample 2019 | 4.40 | 25.56 | 4.72 |
| Groundwater sample 2021 | 4.77 | 26.25 | 4.67 |
| Groundwater sample 2022 | 4.69 | 27.05 | 4.61 |

In Figure 6 the range of trace element to calcium ratios that could occur in calcite precipitated from the measured groundwater are presented (shaded bands) alongside trace element to calcium ratios observed in the speleothems (box and whisker plots).
It is worth noting that only one set of partition coefficients was calculated from subaqueously precipitated calcite (Drysdale et al., 2019), and none from calcite rafts (which precipitate on the surface of the water body, not at depth).

## 4.3 Groundwater calcite saturation

Measurements of the groundwater sample taken in 2022 (Table 2) were used to calculate the calcite Saturation Index (1.12).

**Table 2 Physical properties and major anions measured on the groundwater sample collected in 2022.**

| | |
|---|---|
| Temperature (°C) | 21.0 |
| pH | 8.11 |
| Total alkalinity (mg $CaCO_3$/L) | 294 |
| Bicarbonate alkalinity (mg $CaCO_3$/L) | 291 |
| Carbonate alkalinity (mg $CaCO_3$/L) | 3 |
| $Cl^-$ (mg/L) | 940 |
| $SO_4^{2-}$ (mg/L) | 262 |
| $Br^-$ (mg/L) | 3.39 |
| $NO_3^-$ (mg/L) | 10.7 |





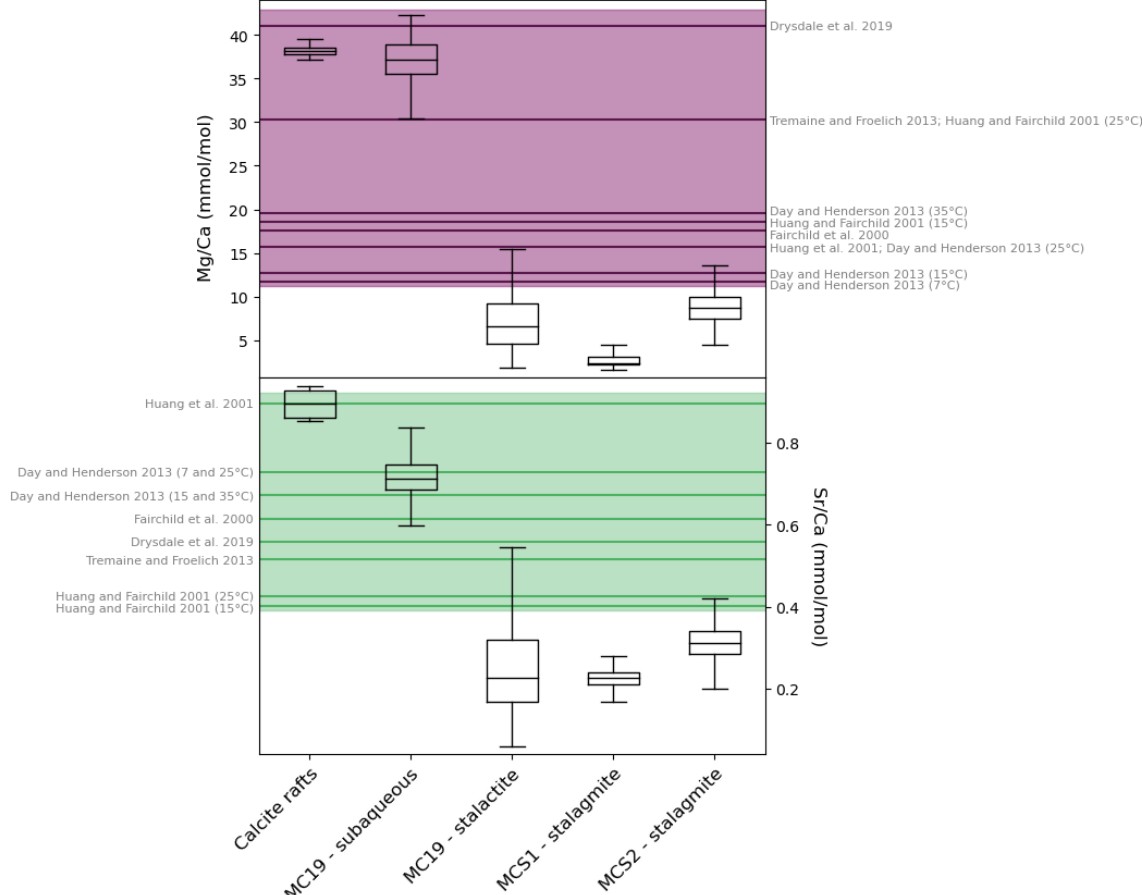

**Figure 6   Upper panel boxplots: Mg/Ca ratios in cave calcites. Lower panel boxplots: Sr/Ca ratios in cave calcites. The blue (upper) and orange (lower) lines indicate the X/Ca ratios that could occur in calcites precipitated from the groundwater. These values were calculated using the mean X/Ca ratios measured in the groundwater. The partition coefficients for each individual calculation are indicated in grey text. The blue (upper) and orange (lower) shaded areas represent the full range of X/Ca ratios that could occur given the full range of X/Ca ratios measured in the groundwater.**


### 4.4 Speleothem age

The positions of the U-Th samples are indicated by orange lines in Figure 4. Two samples taken from the base of the embedded stalactite within the pendulite returned ages beyond the limits of uranium-thorium dating, with isotopic ratios indicating that the stalactite began growing more than 702 kyr (MC19-18 and -13 in Table 3). Three other laminae within the stalactite

returned ages of 513.0 (+85/-50), 258.1 (± 4.6) and 250.0 (± 4.7) kyr (MC19-19, -15, and -14 in Table 3). The transition to subaqueous growth appears to have occurred around 68.5 (± 0.40) kyr (MC19-02 in Table 3) and continued until 65.4 (± 0.49)





kyr (MC19-4 in Table 3). This growth period was followed by a 14-kyr hiatus, with subaqueous growth recommencing at 51.2 (± 0.38) kyr (MC19-05 in Table 3) and continuing through to 42.3 (± 0.28) kyr (MC19-12 in Table 3). A further brief burst of subaqueous growth occurred around 18.9 (± 0.23) kyr (MC19-17 in Table 3). A sample taken from the external growth lamina

returned an age of 16.4 (± 0.20) kyr (MC19-20 in Table 3). Less than 1 mm of calcite was deposited during each of these last two periods of cave flooding. This is insufficient material for more than one dating sample so the duration of these younger growth phases cannot be constrained. However, given the average growth rate during the two longer phases of subaqueous growth (~ 0.7 mm/kyr), the 1 mm of calcite deposited during these two brief bursts could correspond to ~ 1.4 kyr of growth. Initial uranium activity ratios ($[^{234}U/^{238}U]_i$) are nearly four times lower in the stalactite samples than in those from the

subaqueous overgrowth.

All calcite rafts returned ages within the last millennium. Rafts would have undoubtably formed during older cave flooding events that may have coincided with pendulite growth, however, their fragile nature would make long-term preservation unlikely, especially in light of recreational caving, archaeological excavation, and mining activities that have occurred since

at least the 1920s. The dusty calcite raft (MCCR-33_1, 2, 3, 4 in Table 3) produced ages of 1750, 1560, 1650, and 1710 CE, with ± 2s uncertainties of 1200, 260, 200 and 140 years respectively. Assuming coevality between subsamples, an initial thorium calculation was made ([2.41 ± 0.24]) and ages of all rafts corrected accordingly. Although the uncertainties decrease with decreasing dust contamination, all are substantially higher than the uncertainties on the ages of the clean rafts, which average ± 8 years. One of the clean calcite rafts formed in 1073 (± 18 yr) CE (MCCR-29 in Table 3), whilst the remaining

clean rafts were dated to between 1879 and 1968 CE.

**Table 3** U-Th dating results for pendulite MC19 and calcite rafts. Square brackets indicate activity ratios. Numbers in brackets are 2σ uncertainties of the least significant digits.

| Sample ID | Depth mm (±) | U ng/g | $[^{230}Th/^{232}Th]$ | $[^{230}Th/^{238}U]$ (2σ) | $[^{234}U/^{238}U]_i$ (2σ) | $[^{232}Th/^{238}U]$ (2σ) | Uncorr. age kyr (2σ) | Corr. age kyr (2σ) | $[^{234}U/^{238}U]_i$ corr. (2σ) |
|---|---|---|---|---|---|---|---|---|---|
| Pendulite (stalactite interior) | | | | | | | | | |
| MC19-18 | | 1258 | 299.5 | 1.2490(46) | 1.1581(18) | 0.004170(83) | | No age | |
| MC19-13 | | 267 | 891.5 | 1.2235(80) | 1.1577(34) | 0.001372(27) | 705.05(264.54) | 702(+inf/-149) | 2.2823(2.7950) |
| MC19-19 | | 299 | 3787.0 | 1.1922(69) | 1.1448(31) | 0.000315(06) | 518.11(67.09) | 513(+85/-50) | 1.6275(1182) |
| MC19-15 | | 573 | 146.5 | 1.3983(63) | 1.4215(28) | 0.009546(191) | 258.24(4.47) | 258.1(4.6) | 1.8735(99) |
| MC19-14 | | 434 | 1087.0 | 1.2691(64) | 1.3219(29) | 0.001167(23) | 250.01(4.71) | 250.0(4.7) | 1.6518(79) |
| Pendulite (subaqueous portion) | | | | | | | | | |
| MC19-02 | 19.4(0.8) | 2227 | 1963.5 | 1.8314(79) | 3.6786(54) | 0.000933(19) | 68.48(0.40) | 68.42(0.40) | 4.2497(63) |
| MC19-03 | 18.1(1.0) | 2086 | 2661.5 | 1.8988(86) | 3.8646(59) | 0.000713(14) | 67.26(0.41) | 67.19(0.41) | 4.4633(69) |
| MC19-04 | 16.1(0.7) | 2051 | 3266.4 | 1.8884(106) | 3.9279(75) | 0.000578(12) | 65.39(0.49) | 65.32(0.49) | 4.5213(88) |
| MC19-05 | 13.1(0.9) | 2003 | 1586.7 | 1.5104(88) | 3.8244(70) | 0.000952(19) | 51.31(0.38) | 51.23(0.38) | 4.2645(79) |
| MC19-06 | 12.1(0.8) | 1637 | 1527.6 | 1.4752(79) | 3.8599(64) | 0.000966(19) | 49.31(0.33) | 49.25(0.34) | 4.2869(71) |
| MC19-07 | 9.9(0.5) | 1513 | 3133.6 | 1.4774(92) | 3.9537(76) | 0.000471(09) | 48.01(0.37) | 47.94(0.37) | 4.3822(84) |
| MC19-08 | 8.5(0.7) | 1543 | 3334.6 | 1.4804(81) | 3.9789(67) | 0.000444(09) | 47.75(0.32) | 47.69(0.32) | 4.4086(73) |
| MC19-09 | 5.5(0.7) | 2226 | 1967.5 | 1.4073(72) | 3.9613(61) | 0.000715(14) | 45.21(0.28) | 45.14(0.28) | 4.3643(67) |
| MC19-10 | 4.5(0.8) | 1939 | 2568.1 | 1.3967(80) | 3.9797(69) | 0.000544(11) | 44.57(0.31) | 44.50(0.32) | 4.3791(75) |
| MC19-11 | 3.2(0.9) | 2242 | 593.3 | 1.3861(75) | 4.0053(65) | 0.002336(47) | 43.77(0.31) | 43.70(0.31) | 4.4004(71) |
| MC19-12 | 2.1(0.8) | 2026 | 750.6 | 1.3567(72) | 4.0361(64) | 0.001807(36) | 42.32(0.28) | 42.25(0.28) | 4.4212(70) |




| | | | | | | | | | |
|---|---|---|---|---|---|---|---|---|---|
| MC19-17 | 1.0(0.4) | 1406 | 151.4 | 0.6702(50) | 4.0744(68) | 0.004426(89) | 19.00 (0.23) | 18.94(0.23) | 4.2438(72) |
| MC19-20 | 0.2(0.1) | 1404 | 132.9 | 0.5846(34) | 4.0693(53) | 0.004398(88) | 16.43(0.20) | 16.36(0.20) | 4.2149(57) |
| Dusty calcite raft | | | | | | | | | |
| MCCR-33_1 | | 2842 | 6.2 | 0.0040(03) | 4.0909(168) | 0.000679(45) | 0.080(28) | 0.007(0.028) | 4.0916(168) |
| MCCR-33_2 | | 2949 | 10.8 | 0.0033(04) | 4.0850(264) | 0.000338(32) | 0.075(17) | 0.002(0.017) | 4.0857(264) |
| MCCR-33_3 | | 2809 | 11.4 | 0.0038(01) | 4.0721(200) | 0.000344(26) | 0.088(14) | 0.015(0.014) | 4.0728(200) |
| MCCR-33_4 | | 2694 | 12.8 | 0.0059(02) | 4.0739(199) | 0.000473(33) | 0.139(20) | 0.066(0.020) | 4.0751(199) |
| Clean calcite rafts | | | | | | | | | |
| MCCR-10 | | 2554 | 7.4 | 0.0031(03) | 4.0746(79) | 0.000432(12) | 0.083(08) | -0.0179(0.0086) | 4.0751(79) |
| MCCR-12 | | 2514 | 11.0 | 0.0040(02) | 4.0806(91) | 0.000372(10) | 0.107(05) | 0.0100(0.0060) | 4.0813(91) |
| MCCR-16 | | 2584 | 9.9 | 0.0056(03) | 4.0712(76) | 0.000573(14) | 0.150(08) | 0.0402(0.0089) | 4.0722(76) |
| MCCR-18 | | 2698 | 12.1 | 0.0059(02) | 4.0773(91) | 0.000498(10) | 0.158(05) | 0.0528(0.0062) | 4.0783(90) |
| MCCR-20 | | 2509 | 15.3 | 0.0041(02) | 4.0780(75) | 0.000274(05) | 0.110(05) | 0.0191(0.0057) | 4.0787(74) |
| MCCR-22 | | 2835 | 8.2 | 0.0077(02) | 4.0810(80) | 0.000957(19) | 0.206(05) | 0.0713(0.0083) | 4.0823(80) |
| MCCR-23 | | 2230 | 3.0 | 0.0305(05) | 4.0626(78) | 0.010261(264) | 0.821(14) | 0.083(0.070) | 4.0640(78) |
| MCCR-26 | | 2410 | 10.0 | 0.0057(03) | 4.0803(86) | 0.000589(21) | 0.152(08) | 0.0415(0.0089) | 4.0813(85) |
| MCCR-25 | | 2561 | 9.5 | 0.0029(02) | 4.0685(86) | 0.000318(06) | 0.078(05) | -0.0158(0.0058) | 4.0690(87) |
| MCCR-29 | | 2430 | 22.0 | 0.0396(05) | 4.0633(70) | 0.001806(30) | 1.067(14) | 0.877(0.018) | 4.0715(70) |
| MCCR-30 | | 4203 | 13.3 | 0.0026(02) | 4.0772(83) | 0.000205(04) | 0.070(05) | -0.0166(0.0055) | 4.0777(83) |
| MCCR-31 | | 2552 | 9.5 | 0.0042(03) | 4.0811(77) | 0.000456(08) | 0.112(08) | 0.0099(0.0086) | 4.0819(78) |

For the purpose of calculating the DCF, four rafts underwent paired U-Th and radiocarbon analyses. Two of the rafts (MCCR-25 and -30) formed during the post-bomb period (1965 and 1966 respectively) when atmospheric $^{14}$C was increasing rapidly. Given the uncertainties on the U-Th ages, atmospheric pMC at the time of raft formation cannot be accurately estimated and these results were consequently discarded. The two rafts (MCCR-26 and -31) that formed in the pre-bomb period (1908 and 1940 respectively) returned DCFs of 41.38 (± 0.31) and 40.45 (± 0.33) %.


**Table 4  Radiocarbon dating results for the calcite rafts and calculated Dead Carbon Fractions.**

| ANSTO code | Sample ID | δ$^{13}$C ‰ | 1σ | pMC | 1σ | yrs BP | 1σ | yrs BP (U-Th) | 2σ | pMC$_{atm}$ | 1σ | DCF (%) | 1σ |
|---|---|---|---|---|---|---|---|---|---|---|---|---|---|
| OZBK55 | MCCR-30 | -5.5 | 0.1 | 58.41 | 0.18 | 4319 | 25 | -16.6 | 5.5 | | | | |
| OZBK53 | MCCR-25 | -6.2 | 0.3 | 57.50 | 0.19 | 4445 | 27 | -15.8 | 5.8 | | | | |
| OZBK56 | MCCR-31 | -5.5 | 0.1 | 58.39 | 0.18 | 4322 | 25 | 9.9 | 8.6 | 98.05 | 0.44 | 40.45 | 0.33 |
| OZBK54 | MCCR-26 | -5.5 | 0.3 | 57.72 | 0.17 | 4415 | 24 | 41.5 | 8.9 | 98.46 | 0.43 | 41.38 | 0.31 |

## 4.5 Groundwater age

The radiocarbon results for the DIC in the groundwater (Table 5) returned a corrected age of 589 cal BP and a DCF of 45.8 %

(Han and Plummer, 2013). The radiocarbon age can be considered the average age of the water within the sample. The presence of tritium (0.13 TU) is evidence that the aquifer also contains a portion of modern (post-1950) recharge (Clark, 2015).





**Table 5  Radiocarbon results for the groundwater.**

| ANSTO code | Sample ID | $\delta^{13}C$ ‰ | $1\sigma$ | pMC | $1\sigma$ | Uncorr. yrs BP | $1\sigma$ | Total $CO_2$ (DIC mEq/L) | Corr. yrs BP |
|---|---|---|---|---|---|---|---|---|---|
| OZAS59 | M1-C1 | -12.9 | 0.6 | 49.92 | 0.14 | 5580 | 25 | 5.76 | 589 |

## 5 Discussion

### 5.1 Mechanism of cave flooding

Together the external morphology, internal stratigraphy, petrography, and trace element profiles of MC19 provide clear evidence for transition from classical stalactitic growth to subaqueous growth. The presence of calcite rafts, a standing water mark, calcite overgrowths on boulders (see Fig. 3 in Site Description), and the evidence from historical documentation over the last two centuries provide further indication that Mairs Cave experiences periodic inundation. This raises the question of the mechanism by which the water enters the cave: invasion by runoff from overland flow, excessive vadose zone infiltration, or a rise in the regional groundwater table?

Buckalowie Creek is an ephemeral waterway that runs parallel to the strike of the Etina limestone formation and the long axis of the cave system (see Fig. 2a in Site Description). The elevation of the modern creek bed is ~10 m lower than the entrance to the cave, but 10 m higher than the lowest point in the main chamber (Fig. 2a and 2c). It is unlikely that the creek could rise 10 m to flood the cave via the entrance shaft given that floodwaters would need to occupy the broad width (400 m) of the floodplain before attaining the level of the entrance shaft. The creek has no nearby tributaries that could divert water into the cave. Similarly, there is no evidence on the land surface above the cave that would indicate flooding has occurred via the entrance shaft. In addition, flooding from overland flow would deliver water that is undersaturated with respect to calcite and would be unlikely to precipitate calcite even once chemical equilibrium was reached with the bedrock margins of the cave. Finally, overland flow would be highly turbid, but speleologists report clear water in the cave during the 1968 and 1974 inundation events. Thus, we eliminate direct invasion of overland flow as the dominant mechanism for cave flooding.

Standing water can be found in caves when periods of excessive drip water discharge causes water to collect in depressions or impounded basins developed in the cave floor (e.g. Drysdale et al. 2019). However, this is only possible where the cave floor is impermeable. The Etina limestone and the neighbouring siltstone formations on either side are all near vertical (Fig. 2), with a consistent dip of ~ 80 degrees along the entire 10 km length of these outcrops. Therefore, it is unlikely that there is an aquitard below the cave that could allow water to pool to create a perched aquifer. Further, an impossibly high discharge of vadose percolation water would be required to fill the volume of the main chamber to the observed level of the standing water mark, or the water level observed by cavers in 1974.




The calcite rafts and the subaqueous portion of the pendulite exhibit similar Mg/Ca and Sr/Ca values, which are 2-4 times higher than the values observed in the subaerial speleothems i.e. the stalactite portion of the pendulite and the two stalagmites
studied by Treble et al. (2017). This confirms that the subaqueous and the subaerial (drip-fed) calcites precipitated from different parent waters, and that pooling drip water is unlikely to be the parent water from which the subaqueous formations precipitated. Rising of the regional water table is therefore the most likely mechanism to explain the flooding of Mairs Cave. The groundwater is supersaturated with respect to calcite ($SI_{cc}$ = 1.12) meaning calcite will readily precipitate from the groundwater. The Mg/Ca and Sr/Ca values of the subaerial calcites fall outside of the ranges of values that could occur in
calcites precipitated from the groundwater (see Fig. 6 in Results). The Mg/Ca and Sr/Ca values of the subaqueous calcites fall within these ranges, indicating that it is entirely possible for the subaqueous formations to have precipitated from the groundwater.

U-Th dating revealed markedly different initial uranium isotope ratios ($[^{234}U/^{238}U]_i$) between the subaerial calcite (1.56 to 2.95)
and the subaqueous calcite (4.07 to 4.52). Alpha-recoil during radioactive decay of $^{238}U$ in the bedrock causes $^{234}Th$ to be expelled across grain boundaries and into pore spaces where it rapidly decays into $^{234}U$ and can be entrained by infiltrating water (Kigoshi, 1971; Osmond et al., 1992; Osmond and Cowart, 1976; Plater et al., 1992). $^{234}U$ can also preferentially diffuse into pore water via alpha recoil tracks. The longer that infiltrating water spends in the bedrock the more $^{234}U$ it will acquire. This would explain why subaqueous calcites have higher $[^{234}U/^{238}U]_i$ than the calcites precipitated from dripwater, considering
some studies have observed dripwater arriving in a cave in a matter of hours (Markowska et al., 2016), while the groundwater has a residence time of 661 years, according to the radiocarbon dating results. Therefore, the initial uranium isotope data also indicates rising groundwater is the mechanism responsible for flooding Mairs Cave.

Meteoric water that infiltrates the subsurface becomes slightly acidic by dissolution of $CO_2$ that is present in the soil due to
microbial and root respiration (Fairchild & Baker, 2012). The $[^{14}C]$ of the soil $CO_2$ should be close to or at equilibrium with the atmosphere (Fohlmeister et al., 2011). When the water subsequently percolates through and dissolves carbonate bedrock the $[^{14}C]$ of the DIC will be reduced because carbonate bedrocks, typically on the order of $10^5$-$10^8$ years old, exhibit a $[^{14}C]$ equal to zero (Clark, 2015). The DCF is a term used to describe what portion of the carbon present in a sample was contributed by exchange with the bedrock or other 'dead' carbon reservoirs. The groundwater in Buckalowie Valley returned a DCF of
45.8 %, meanwhile the calcite rafts returned DCFs of 41.38 and 40.45 %. Considering uncertainties inherit to radiocarbon corrections, and the DCF variability observed in speleothems (Griffiths et al., 2012; Hua et al., 2012; Welte et al., 2021) the DCFs of the groundwater and the calcite rafts show remarkable similarity, which serves as additional evidence that flooding of the cave can be attributed to rising of the water table during periods of enhanced groundwater recharge.





## 5.2 The groundwater context

Assuming similarity to other groundwater systems in the Ikara-Flinders Ranges, the impermeable formations that comprise the valley walls most likely act as hydrological divides that constrain the extent of the aquifer (Ahmed et al., 2021; Ahmed & Clark, 2016; Fildes et al., 2020). However, given the high degree of fracturing caused by the orogenic history of the region it is highly likely that the aquifer hosted in the limestone exhibits some hydrological connectivity to neighbouring groundwater systems. Nevertheless, the age (588.84 years) and shallowness (5.54 m Depth to Water) of the groundwater in the Buckalowie valley are indicative of a localised and dynamic system that is unlikely to be mixing with a deeper and older regional aquifer.

The fact that the groundwater sample contains a portion of modern water (indicated by the presence of tritium) suggests that the bore and cave are within the recharge region of an active groundwater system, implying no significant lag between when meteoric waters first enter the subsurface and their arrival in the cave. For example, if Mairs Cave were to experience a prolonged rise in the water table to (re-)initiate subaqueous growth, the 'age' of the infiltrating waters submerging the pendulites would likely be relatively young ($10^0$-$10^1$ years) or at most contain a component of $10^2$ years old water. Thus, we interpret that millennial-scale climate events should be detectable in a proxy record from MC19, with a temporal offset of no more than $10^1$-$10^2$ years, assuming groundwater flow has not changed significantly since subaqueous pendulite growth began. Considering there does not appear to be any significant mixing with older groundwaters, proxy signals are unlikely to exhibit excessive temporal smoothing.

The groundwater context in which MC19 grew offers advantages over classical (vadose zone drip-fed) speleothems. Monitoring studies have shown that dripwater chemistry may be heterogenous between drip points, due to heterogeneity in flow paths and/or geochemical composition of the bedrock. The effects of these factors are often amplified in arid regions (Fairchild et al., 2006; McDonough et al., 2016) and can translate to coeval stalagmites returning dissimilar geochemical records from the same cave, raising the need for duplication of stalagmite records and/or interpretations that are supported by cave monitoring (Treble et al., 2022). In contrast, the proxy signals that MC19 inherits from the groundwater will be homogeneous on the local scale and the rise and fall of the water table is more likely to be responsive to regional hydroclimate variability rather than the nuances of a single drip path.

Finally, 'Thorium scavenging' has been identified as a potential issue with the use of subaqueous speleothems for palaeoclimate reconstruction (Drysdale et al., 2019, 2020; Moseley et al., 2016). $^{230}$Th present in the water due to the radioactive decay of dissolved uranium can be 'scavenged' by speleothem calcite. The addition of non-authigenic $^{230}$Th causes the calcite to appear older than it is. Where $^{230}$Th concentration increases with depth, samples taken from deeper in the water column will appear systematically older than those taken from higher in the water column (Moseley et al., 2016). In a setting where water depth fluctuates over time, the speleothem will be exposed to variable $^{230}$Th concentrations, resulting in age



inversions. Despite fluctuating water levels in Mairs Cave, all ages taken from MC19 are in stratigraphic order, which suggests that thorium scavenging is unlikely to have altered the isotope activity ratios in the calcite.

**5.3 Palaeoenvironmental interpretations**

The calcite rafts in Mairs Cave are indisputable evidence that cave flooding has occurred since they can only form at the surface of a body of water. The three youngest calcite rafts, dated to 1966 (± 5.8 years), 1967 (± 5.5 years) and 1968 (± 8.6 years), most likely correspond to the cave flooding event reported by cavers in 1968. Unfortunately, the age uncertainties are such that beyond the historical record the rafts cannot be used to confidently identify individual flooding events. While cave
flooding during the last millennium was responsible for the precipitation of many calcite rafts, no appreciable amount of calcite was deposited on the surface of the pendulite, indicating that cave flooding has been sporadic and ephemeral. Studies of groundwater systems in arid and semi-arid regions are increasingly showing that groundwater recharge occurs only when a site-specific infiltration threshold is exceeded by an extreme rainfall event (Boas and Mallants, 2022; Gelsinari et al., 2024; Jasechko, 2019; Jasechko and Taylor, 2015; Taylor et al., 2013; Villeneuve et al., 2015). At Mairs Cave extreme rainfall events
occur more often in summer than in any other season (BoM, 2024a). During summer the Subtropical Ridge sits at ~ 37 ° (Larsen and Nicholls, 2009), preventing rainfall coming off the Southern Ocean from reaching Mairs Cave (32 °S), hence rainfall events during summer are primarily a consequence of southward propagation of IASM depressions. Cave flooding in 1974 coincided with continent-wide rainfall extremes caused by an intensification and southward displacement of the IASM (Pook et al., 2014; BoM, 2024b).


Dates from the two most exterior laminae of MC19 returned ages of 16.4 (± 0.20) kyr and 18.9 (± 0.23) kyr (Fig. 7A). We judge these to be two distinct bursts of growth rather than a period of continuous growth because the two laminae are separated by a micritic/microsparitic layer. The combined growth span of stalagmites MCS1 and MCS2 is from 24 to 15 kyr (Fig. 7A), but as interpreted from the isotopic records, the period of highest recharge to the groundwater occurred between 18.9 and 15.8
kyr (Treble et al., 2017), showing close alignment with the two youngest growth bursts in the pendulite. A minimum in $\delta^{18}O$ at 17.2 ± 0.08 kyr was interpreted by Treble et al. (2017) as an intensification of the IASM triggered by Heinrich 1 (H1; 17.5 to 14.7 kyr; El Bani Altuna et al., 2024). The burst of subaqueous growth in MC19 at 16.4 (± 0.2) kyr may also correspond to this event. Intensification and southward migration of the IASM due to H1 is evident in marine sediment records from Indonesia (Ardi et al., 2020; Muller et al., 2012; Steinke et al., 2014), and speleothem records from northern Australia
(Denniston et al., 2013a, c, b) and Indonesia (Ayliffe et al., 2013). Climate model precipitation outputs for the semi-arid and arid regions of Australia demonstrate a period of positive moisture balance between 17.5 and 14.7 kyr which is attributed to increased delivery of tropical rainfall (Cadd et al., 2024). Given the broader continental context, it is plausible that H1 could have been responsible for the pulse of pendulite growth in Mairs Cave at 16.4 (± 0.20) kyr via southward displacement and/or intensification of the IASM.






The two extended phases of subaqueous calcite deposition on MC19 reveal that during the LGP cave flooding and thus enhanced groundwater recharge occurred not just as sporadic and ephemeral bursts but was sustained over millennia. This is indicative of a much more positive water balance than in the present day. These findings support an emerging paradigm shift which suggests that, in an Australian context, moisture availability is elevated during glacials relative to interglacials (Weij et al., 2024). Flooding of Mairs Cave sustained over multi-millennial periods during the LGP could reflect more frequent southward incursions and/or increased intensity of the IASM, in conjunction with a reduced recharge threshold due to lower evapotranspiration. This interpretation is consistent with findings from the Naracoorte Cave Complex (600 km SSE of Mairs Cave) where speleothem growth under glacial climates has been linked to tropically sourced, warm-season moisture (Weij et al., 2024).

The most recent multi-millennial growth period in MC19 occurred between 51.2 (± 0.38) and 42.3 (± 0.28; Fig. 7A). We propose that this period of pendulite growth can also be attributed to the IASM since (1) there is SH summer insolation maximum at 46 kyr (Fig. 7B; Laskar et al., 2004), (2) the most recent period of interconnectivity between Kati Thanda and Munda occurred between 49.4 (± 2.4 kyr) and 46.8 (± 2.3) kyr (Cohen et al., 2015), (3) high lake levels were sustained between ~50 and 37 kyr at Paruku-Gregory Lakes (20 °S; Bowler et al., 2001; Fitzsimmons et al., 2012; Veth et al., 2009), (4) a highstand at Lake Woods (18 °S) between 53 and 30 kyr was proposed by Bowler et al. (1998), (5) neodymium isotope data from MD03-2607 (Fig. 7E) suggests an increase in outflow from the Darling River between 48 and 46 kyr (Bayon et al., 2017), and (6) from 60 to 40 kyr low percentages of subpolar and transitional foraminifera species in ocean core MD03-2611 (Fig. 7F; 36 °S) would suggest that the sub-tropical and sub-polar fronts were positioned further south relative to the present (De Deckker et al., 2020). Under these conditions southern Australia would be unlikely to be receiving any significant amount of rainfall off the Southern Ocean, indicating that pendulite growth between 51.2 and 42.3 kyr was also likely related to increased moisture delivery from the IASM.

The cause of the older phase of pendulite growth (68.5 to 65.4 kyr) is more difficult to designate. Regressional facies dated to 70.8 (± 3.8) kyr suggest a decline in Kati Thanda lake levels (Fig. 7c) that may have persisted until 60 (± 2) kyr, to when the next highstand is dated (Cohen et al., 2022). This decline is not evident in Munda (Fig. 7d), where high lake levels were sustained between 68.2 (± 4.2) kyr and 60.1 (± 7.6) kyr (Cohen et al., 2015). A rapid and dramatic increase in the subpolar foraminifera species in MD03-2611 (Fig. 7f) between 70 and 60 kyr suggests a northward displacement of the subpolar front (De Deckker et al., 2020). Meanwhile, neodymium isotope data from MD03-2607 (Fig. 7e) suggests a southward shift in the IASM between 68 and 54 kyr (Bayon et al., 2017), and there is an increase in speleothem growth at Naracoorte that appears closely coupled to the peak in local summer insolation between 75 and 65 kyr (Weij et al., 2024). These contradictory results make it difficult to elucidate the behaviour of the IASM and the SHWWs throughout this period, and further interrogation of the pendulite record will be required to identify the cause of this growth phase.





**Figure 7** **Shaded vertical bars indicate when Mairs Cave speleothems were growing. (a) Dates (including 2σ error bars) from Mairs Cave speleothems. (b) Summer insolation at 32.5 °S from Laskar et al. 2004. (c) Height of shorelines at Kati Thanda-Lake Eyre. Filled diamonds are data from Magee et al. 2004. Unfilled diamonds are data from Cohen et al. 2015, 2022. (d) Height of shorelines at Munda-Lake Frome. Filled diamonds are from Cohen et al. 2011, 2012. Unfilled diamonds are from May et al. 2015. Unfilled circles are from Gliganic et al., 2014 (e) Neodymium isotopes (ε Nd) in MD03-2607 representing relative sediment contributions from Darling River (− 2.4 ± 2.4) and River Murray (− 9.5 ± 0.9) from Bayon et al. (2017). An increase in contributions from the Darling River is considered indicative of an increase in the strength of the IASM. (f) Percentage of subpolar planktic foraminiferal species in core MD03-2611 for Murray Canyon from De Deckker et al. 2020. An increase implies northward expansion of the subpolar and subtropical fronts, most likely accompanied by the SHWWs.**




## 6 Conclusion

Observations within the cave, as well as historical archives and caving reports, suggest that Mairs Cave is periodically flooded. Speleothems hanging from the cave ceiling exhibit external and internal morphology typical of pendulites, which form when stalactites become submerged in water that is saturated with respect to calcium carbonate. Subsequent growth continues

subaqueously whilst the pendulite is submerged but ceases when the water recedes.

Due to the geology and geomorphology of the landscape surrounding the cave, overland flow can be ruled out as the cause of cave flooding. Trace element to calcium ratios in the subaerial calcite differ greatly from ratios in the subaqueous calcite, suggesting they precipitated from different parent waters, ruling out pooling dripwater as the cause of cave flooding. Given

the calcite saturation of the groundwater ($SI_{cc}$ = 1.12) calcite would be readily precipitated. The trace element to calcium ratios in the subaqueous calcite fall within the range of values that could theoretically occur in calcite precipitated from the measured groundwater. There is remarkable agreement between the DCFs of the calcite rafts (41.38 and 40.45 %) and the groundwater (45.8 %) and the subaqueous calcite exhibit much higher initial uranium isotope ratios than the subaerial calcites ($\mu$ = 4.00 versus $\mu$ = 1.78). All lines of evidence support the conclusion that rising of the water table during periods of enhanced

groundwater recharge is the cause of cave flooding events. Consequently, we interpret that pendulite growth indicates periods when the water table was higher relative to the present.

The presence of tritium (0.13 TU) in the groundwater sample indicates that the bore and cave are within an active recharge region, and the $^{14}$C age (589 yrs BP) indicates negligible mixing with a deeper and older regional aquifer. Therefore,

geochemical signals contained in the subaqueous speleothems are unlikely to exhibit smoothing or a temporal offset $\geq 10^2$ years. Dating of the pendulite subaqueous overgrowth returned ages in stratigraphic order, so it appears unlikely that thorium scavenging has occurred. We therefore conclude that the subaqueous pendulite growth may be ideal for further paleoclimate investigations as it is a record of past water table high stands, which are a function of groundwater recharge and hydroclimate variability.


Historical observations, the ages of the calcite rafts, and the lack of pendulite growth indicate that strong water deficits under warm Holocene interglacial conditions give rise to episodic, rather than persistent, cave flooding. Emerging research indicates that only extreme rainfall events are capable of triggering groundwater recharge in arid settings; at Mairs Cave extreme rainfall events are largely delivered during the summer months by the IASM. During the LGP, the pendulite underwent two multi-

millennial growth phases and two short bursts of growth. The youngest burst of growth (16.4 kyr) corresponds to an intensification and southward displacement of the IASM triggered by H1. Alignment of pendulite growth phases with maxima

in local summer insolation would suggest that increased groundwater recharge was caused by increased southward displacement and/or intensity of the IASM. Comparison to other palaeoclimate records largely confirms that the longest of the two pendulite growth phases (51.2 to 42.3 kyr) was a response to the behaviour of the IASM. Contradictory findings across
other Australian records mean the cause of the older pendulite growth phase (68.5 to 65.4 kyr) cannot yet be confidently attributed to either the IASM or the SHWWs.

While the cause of Australia's pluvial periods through the LGP will require further investigation, the pendulite from Mairs cave offers a clear advantage over other archives from the arid zone; the timing of pluvial periods can be constrained with
much greater precision. Situated on the southern boundary of Australia's arid zone, and at the interface between mid-latitudinal and tropical climate systems, Mairs Cave will undoubtably provide much needed insight into the climate of a region wanting in paleoclimate records. Future studies will seek to reconstruct the climate of Australia's arid zone with greater temporal resolution using proxy records from multiple pendulites.

**Author contributions.**

CGW, RD, PT and JHM designed the project and wrote the paper, with contributions from all authors. CGW, RD, JH, and CB collected the calcite and groundwater samples. CGW milled powdered samples, performed petrographic analysis and prepared samples for U-Th dating. JH performed U-Th dating. PT calculated the calcite saturation index of the groundwater. SP corrected the groundwater radiocarbon results and advised on the interpretation of all groundwater results. JHM advised on the comparison to regional palaeoenvironmental archives. CB sourced historical documents and caving records.

**Competing interests.**

At least one of the (co-)authors is a member of the editorial board of Climate of the Past.

**Acknowledgements.**

We acknowledge the support for the Centre for Accelerator Science at ANSTO through the Australian National Collaborative Research Infrastructure Strategy (NCRIS). We are grateful to Quan Hua for performing the radiocarbon analyses on the calcite
samples and assisting in the interpretation of the results. We thank Alan Greig for performing the trace element analysis by LA-ICP-MS. We thank Henri Wong and Chris Vardanega for performing trace element analyses of calcites and groundwater by ICP-AES, and anion measurements of groundwater by IC. We thank Silvia Frisia for her contributions to the classification of calcite fabrics.



**Financial support.**

This project was funded by Australian Research Council Discovery Project grant DP220102134 (to RD, PT and JH) and
Herman Slade Foundation grant HSF21073 (to RD, CGW and CB). We acknowledge the support for the Centre for
Accelerator Science at ANSTO through the Australian National Collaborative Research Infrastructure Strategy (NCRIS).
CGW was the recipient of an Australian Institute of Nuclear Science and Engineering (AINSE) Post-Graduate Research Award
(PGRA) which funded analyses performed at ANSTO.

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
