# Peer review of "Subaqueous speleothems as archives of groundwater recharge on Australia's southern arid margin"

_EGUsphere, 2024_

## Author Response (AR1)

**Author's Response**

Referee 1:

This paper presented detailed studies of a subaqueous speleothem from Mairs Cave in South Australia. By systematically analyzing the trace element concentrations, uranium isotope ratios, and Dead Carbon Fractions, the authors suggested that the enhanced groundwater recharge was the cause of the cave flooding events that trigger the pendulite growth. They indicated two multi-millennial growth phases (68.5 to 65.4 kyr and 51.2 to 42.3 kyr) and two short bursts of growth (18.9 kyr and 16.4 kyr) during the Last Glacial Period in this region, which corresponding to maximum local summer insolation.

I read this paper with great interest, because few studies were done in pendulites when comparing to stalagmites. In addition, speleothem records are very important for understanding the past hydroclimate change of this arid region. Therefore, I recommend its publication with minor revisions.

1. Line 13, "As anthropogenic climate change enhances aridity across vast regions of the globe,"I think this is a debated question, not a conclusion. Changed 'vast' to 'many'.

2. Line 38-42, the descriptions of aeolian and paleolacustrine depositions in arid region are too arbitrary. Some of them well well studied and have very good climate presentations in arid regions in the world. We have adjusted to text to emphasise the comparison between the archives available in Australia's arid zone, versus those available along the more humid coastal fringes of the continent.

3. I suggest to reduce the conclusion part and make it more concise. Conclusion has been reduced.

4. The colors of different dates in Fig 7a are not easy to distinguish, you may consider to change to some striking colors. The colours were taken from the categorical palette 'hawaii' in Crameri et al. 2020, to improve readability for those with colour-vision deficiency. This citation has been added to the manuscript in the Figure captions.

Referee 2:

The manuscript by Gould-Whaley et al. presents trace element and cation concentration data, alongside U/Th and radiocarbon ages of various speleothems and water samples. The authors have also calculated the dead carbon fraction, contributing valuable insights into groundwater recharge dynamics in a region with limited paleoclimatic proxies. Overall, this is a well-crafted manuscript with sound results, and it is a meaningful addition to research from Australia's southern arid margin.

My review is divided into two parts: the first addresses the terminology used, which I consider an important aspect, and the second includes comments on other elements that may improve the manuscript with a straightforward revision.

**1. Specific comment**

I was surprised by the use of the term "pendulites." This term appears to have previously been applied specifically to partially submerged stalactites in the Jewel and Lechuguilla caves. Identical speleothem forms have been known for decades under various names, such as "war-cup," "war-club-shaped stalactites," and "drumstick." While "pendulites" is included in a Cave and Karst Lexicon published in 2002, it remains relatively uncommon in the scientific literature on caves.

After examining the references provided by the authors, I found that these primarily refer to a specific type of cave deposit known as "phreatic overgrowths on speleothems" (POS), which have been documented in caves in Mallorca, Sardinia, and Cuba. These POS deposits form in brackish water environments in caves located within a few hundred meters of the coastline, with their morphology controlled by tidal ranges. POS are used in several high-profile studies to reconstruct sea-level changes across Holocene, MIS 5and Pliocene. Moreover, the POS formations referenced in these papers are shown to form on stalactites, stalagmites, columns, or directly on cave walls, not only on stalactites as "pendulites".

Given this, citing these references could imply that "pendulites" and POS are identical in form and genetic mechanism; however, based on my reading, they are distinct. While both forms develop subaqueously, the factors that control their morphology and their significance (sea level vs. groundwater recharge) differ substantially. I recommend that the authors clarify this to avoid potential confusion. Specifically, it would be helpful for the manuscript to either distinctly differentiate between "pendulites" and POS by providing a parallel description or define "pendulites" without referring to POS.

The pendulite in our study is comprised of material deposited subaqueously on an existing speleothem within the phreatic zone, which constitutes a Phreatic Overgrowth on a Speleothem (POS). However, as the reviewer points out, studies of POS have thus far focused on coastal caves filled with brackish water, where the height of the phreatic zone is governed by sea level change. As a result, the term POS has become associated with this specific growth context. We originally chose to include the references to POS because they provide detailed descriptions of how variability in the height of the phreatic zone can be reconstructed from discontinuous subaqueous speleothem growth. To our knowledge there have been no such studies of pendulites. To avoid confusion, we have removed references to POS.

**2. Comments/questions/suggestions**

Line 80: Water drips from the ceiling, rather than through the ceiling. Corrected.

Line 101-102: The phrase "seal off the original path of drip" may be unnecessary. Once a speleothem is submerged, gravitational growth cannot continue, as water cannot flow through the internal soda straw. Removed.

Line 116: Please provide the Köppen climatic classification for better clarity. Done.

Lines 145-148: Including a panoramic view or a series of images from this room would help readers assess the positions of the subaqueous overgrowth (pendulites) in relation to coated

breakdown and wall coatings. In Figure 3, it is challenging to identify water levels responsible for calcite precipitation. Such images could be added in a supplementary file. The position of the pendulites in relation to the other features mentioned can be read from Figure 2. We believe it is important to include the photos from the cave (Figure 3) as they visually communicate what we describe in the text.

Line 171: In Figure 2, please clarify the abbreviation "mAHD". Done.

Line 195: The phrase "curtain of bulbous formations" could be clarified. Are these formations the "pendulites"? If so, consider using this term or describing them as "stalactites with subaqueous overgrowth." Trimmed to just use the term 'pendulites'.

Line 201: Consider using "pendulite" or "subaqueous overgrowth" for precision, as speleothem is a more general term. Changed to 'pendulite'

Line 203-204: Not clear. Was this only one thin section from the entire speleothem or one from a certain area of it? Maybe you can show its position in Figure 4. Done.

Additionally, "deposition state" could benefit from further clarification. Done.

Line 214: Please verify the spot size values, as they seem large for a pre-ablation spot size. Are these values in nm or μm? Corrected.

Line 215-216: Use superscript formatting where appropriated. Corrected.

Line 243-243: The sentence mentions "seven clean calcite rafts (MCCR-10, 16, 18, 20, 22, 29)," but I count six samples. Please double-check for accuracy. Corrected.

Line 279-281: Do you really need all these references for a calibration curve? Removed all but Plummer and Glynn, 2013.

Line 294-296: The text discusses a "pendulite," but cites a study (Boop et al.) that describes the genetic origin of phreatic overgrowths on speleothems. Consider addressing this to clarify the distinction. POS citation replaced with pendulite citations.

Line 314: Figure 5: Please indicate from which part of Fig. 4 the thin section was made. Done.

Line 319: In Figure 4 there is a purple line. I do not see a yellow line. Corrected.

Line 335: Table 2. It would be helpful to use the same units as in Table 1 (mmol/L or μmol/L). Since alkalinity is typically reported in mg $CaCO_3$/L, we chose to report the anions with the same units to maintain consistency within Table 2.

3 should be subscript in $CaCO_3$ Corrected.

Figure 6 caption: he caption mentions blue and orange colors that do not appear in the plot. Please verify and correct the caption, as these colors are referenced twice (lines and shaded areas). Corrected.

Line 342: In Figure 4, green lines are visible, though the text references orange. Please verify. Corrected.

Line 366: Table 3: Consider including the $^{232}$Th data to provide a complete dataset. We consider it unnecessary to provide $^{232}$Th content because the values can be calculated from the data we have provided. Absolute Th content has little relevance to the interpretation of U-Th dates and is seldom discussed in that context. Conversely, we do include a redundant column for $^{230}$Th/$^{232}$Th activity because this ratio is widely discussed and understood in the literature as an indicator of detrital Th contamination.

Line 710: "Racovita E." is not listed among the coauthors in the cited study. Please remove this name. Corrected.